# Managing Knowledge in Romanian KIBS during the COVID-19 Pandemic

**Alexandra Zbuchea** *[ID], **Elena Dinu** [ID], **Andra-Nicoleta Iliescu, Roxana-Maria Stăneiu**
and **Bianca-Roxana Salageanu (Șoldan)**

Faculty of Management, National University of Political Studies and Public Administration,
012104 Bucharest, Romania; elena.dinu@facultateademanagement.ro (E.D.);
andra.iliescu@facultateademanagement.ro (A.-N.I.); roxana-maria.staneiu@facultateademanagement.ro (R.-M.S.);
bianca.salageanu.21@drd.snspa.ro (B.-R.S.)
* Correspondence: alexandra.zbuchea@facultateademanagement.ro

**Abstract:** KIBS are increasingly important organizations for ensuring sustainable development. Their core asset is knowledge, manifested in many ways and managed in a complex manner, sometimes jointly with clients. Like other organizations, KIBS companies have been greatly impacted by the COVID-19 pandemic. At the same time, they could provide support to their customers to better cope with the challenges associated with the pandemic. Therefore, the present paper investigates how Romanian KIBS coped with the pandemic by developing 16 interviews with key persons from four different organizations, covering a range of specializations (technical, professional, and creative). The purpose of the present study is to identify the challenges for knowledge management caused by the COVID-19 pandemic and how the pandemic influenced knowledge management performance within Romanian KIBS. The investigation reveals that the pandemic was an opportunity for organizational development and adopting more formal knowledge management practices, as well as for developing the digital profile of companies.

**Keywords:** KIBS; knowledge management in KIBS; impact of the COVID-19 pandemic



## 1. Introduction

Knowledge intensive business services (KIBS) are tightly related to innovation and knowledge. The concept has been proposed by Miles et al. in 1995 [1] but no agreement on its actual significance and coverage has been reached to date. Under a broad view, KIBS are organizations in various sectors such as consultancy and technology (e.g., marketing, legal services, accountancy, software development, scientific research, etc.) that build their services around intensive professional knowledge exploitation. KIBS firms create, accumulate, and disseminate knowledge [1]. KIBS are involved in strong inter-relationships with their clients, which facilitates trust and knowledge transfer. Professional KIBS (p-KIBS), technological KIBS (t-KIBS), and creative KIBS (c-KIBS) tend to act as knowledge and innovation diffusers and facilitators for their clients, or are innovators themselves, driving technological progress [1,2]. A strict and encompassing classification of these companies is very difficult, due to the high complexity and dynamics of this sector. One of the most common classifications used is the structured approach of NACE (the Statistical Classification of Economic Activities in the European Community) [3,4].

KIBS could be at the heart of economic development. Their business models are credited with more vitality compared with other types of businesses, as well as high adaptability [5]. This allows them to offer personalized, dynamic services. Also, the relationships with clients are more complex, based on interaction and co-creation. This leads to a series of challenges, such as effective complex knowledge exchange or dealing with the complexity of internal and external environments [6]. They can positively influence the economy and society in many ways, both at the macro and organizational levels.

The literature in the field has presented many facets of the relationships between KIBS and the socio-economic environment, highlighting the importance of the KIBS sector [7]. They have also been considered integrators for the innovation system because of their mediating position between many types of organizations and stakeholders [8]. Also, innovation spillovers associated with KIBS could be significant, leading to the international competitiveness of countries or business sectors [9].

While, based on data from the National Commerce Register [10], there were over 1,1 million SMEs registered in Romania at the beginning of 2022, no authoritative data or credible sources could be found concerning the number and evolution of KIBS in Romania. Several studies have investigated the profile of the Romanian KIBS sector [11–14], which is an important employer, credited with 8% of the workforce at the national level [11]. The available data shows that p-KIBS are more significant compared to t-KIBS, in terms of numbers and workplaces provided, as well as profits [11]. Also, discrepancies have been documented inside the KIBS sector, with R&D performing lower than other services [12]. At the national level, the KIBS sector has a positive impact on the Romanian economy [11,12,14], as well as on GDP per capita [13]. A high discrepancy between the Bucharest region and the rest of the country has also been documented [11,12,14].

In its country survey, OECD [15] recorded a steady development of the knowledge-intensive service sector in Romania before the pandemic started, with the ICT sector taking the first position. In 2019, for example, 21% of Romanian exports were from ICT services. Nevertheless, the service sector has been affected by the COVID-19 pandemic, and it was already characterized by lower productivity and digitalization. While the regulatory framework to encourage entrepreneurship in professional services should be improved, Romania scores better than the region's average in what concerns barriers to trade and corporate tax rates. Additionally, measures have been taken to support tax exemptions for SMEs, start-ups, and R&D investment [15]. A side effect of the pandemic has been the acceleration of digitalization, as many businesses and services had to switch to the online environment.

Wyszkowska-Kuna [16] conducted a comparative analysis regarding sources of knowledge in manufacturing and service companies in the EU, utilizing data available up to 2016. According to this author, indicators such as the number of professionals employed in manufacturing and services and R&D expenditures in these sectors, as well as the share of KIBS services input into other firms, were comparable in Romania to other countries in Central and Eastern Europe (Poland, Hungary, the Czech Republic, the Slovak Republic, Bulgaria) in the period 2009–2016, with some decrease after the financial crisis of 2007–2009. Another perspective on the Romanian KIBS sector could be associated with the European innovation scoreboard, comparing the innovation performance within the European Union, other European countries, or regional neighbors, with a focus on areas like human resources, attractive research systems, digitalization, finance and support, firm investments, sales impact, and others. According to the 2022 findings, four performance groups have been identified: Innovation leaders, Strong Innovators, Moderate Innovators, and Emerging innovators. Romania's position among the EU countries is in the emerging innovators' group, next to countries like Poland, Slovakia, Bulgaria, Latvia, Hungary, and Croatia [17]. More specifically, Romania has a 32.6% performance ranking compared with the EU countries average and a gap of 17.4% compared with the average performance of countries in its group. Romania's increase is significantly lower than the EU performance increase, with a gap of 9.7% points. Nevertheless, it is important to note that the exports of knowledge-intensive services are one of the relative strengths identified by the extensive study and areas of significant increase since 2021, suggesting that the Romanian KIBS sector is developing, but at a lower rate than the other EU countries.

Studies have presented the positive contributions of KIBS to local and regional development through the complex transfer of knowledge and co-creation [18–22], including new value chains, increased competencies, the diversification of production, and innovative offers [22]. KIBS might act (partially) as a substitute for material capital accumulation,

thereby leading to sustainable development [23]. Cooperation between KIBS and universities is another growth factor, as discussed by several studies [24–26]. KIBS also contribute to increased performance in the networks they are active in [9], including system building.

In terms of the beneficial impact of KIBS on other organizations, especially their clients, the literature has identified a relevant series of benefits, including knowledge transfer and increased innovation [18], audit and diagnosis [27], expert consulting, helping learning processes and problem-solving [18], co-creation [27–30], benchmarking [27], and organizational transformation [31]. All these lead to increased performance and competitiveness.

This is in line with the resource-based view of performance, as documented by Barney [32] and Rumelt [33]. Nevertheless, the literature offers a balanced view between the impact of the industry's characteristics and the resources of a company [34–36]. Knowledge-based theory [37], developed from the classical resource-based theory that considers the main resources of an organization to be financial, material, technological, human, managerial, and relational, and which tends to ignore external factors, stresses the paramount relevance of knowledge, considered its dynamics and interconnectivity both with internal end external environments. In KM, knowledge is the most valuable resource. This is even more relevant in the case of KIBS wherein all inner and external processes are based on knowledge and knowledge dynamics. KIBS organizations benefit from relationships with various stakeholders. The absorptive capacity of KIBS has been investigated, especially in connection with universities and research centers [29]. KIBS firms increase their competitiveness and portfolio following various project and cooperation mechanisms as part of networks of business transactions. Also, strategic alliances are a relevant factor in ensuring a company's performance [38]

Therefore, in times of crises, such as the COVID-19 pandemic, KIBS could be part of the resilience system, but also their agility might help not only themselves but also their network and clients to better cope with the new and significant challenges. These outcomes are connected to an effective knowledge management (KM) approach that KIBS could provide.

The COVID-19 pandemic caused numerous disruptions in the entire ecosystem, but the ones affecting the economic environment generated various types of crises for economies [39–41] and especially businesses [42–44], as well as the financial and banking systems [45–48]. Probably the most consistent approach of economic actors to better face the pandemic has been rethinking organizations to reduce costs, recover revenue streams, rebuild operations, and develop the customer base. Agile management and marketing approaches, as well as accelerated digitalization, have been the most common and effective approaches. To all these, knowledge management is vital.

Considering this framework, the present study explores two research questions:

RQ1. What are the new challenges for knowledge management in KIBS caused by the COVID-19 pandemic?

RQ2. How did the pandemic influence knowledge management performance in KIBS?

Since the process is very complex, the changes induced by the pandemic have been sudden and diverse. A qualitative investigation has been implemented to identify and understand the variety of new situations KIBS companies have been confronted with, knowledge management adjustments brought by the pandemic, and approached solutions. The findings are structured into a section dedicated to KM in Romanian KIBS to better understand the framework, followed by a discussion on the challenges during the pandemic associated with KM processes (both internal and inter-companies) and KM practices, and, finally, an analysis of KM performance in the pandemic context.

## 2. Literature Review

### 2.1. KM for KIBS

Miles et al. [1] state that KIBS companies are characterized by a unique blend of knowledge associated with particular domains, particular applications of technology, and knowledge associated with KIBS' clients. Therefore, complex and diverse knowledge

management is at the core of KIBS development strategies. In the knowledge-based view of the firm, knowledge is a strategic resource for achieving competitive advantage, and its management is paramount for organizational learning [49].

Explicit knowledge, which can be codified and leveraged through the development of an organization's structural capital (organizational systems and infrastructure to leverage intellectual resources), is easily transferable. On the other hand, tacit knowledge (know-how) can be difficult to store and convey. Successful organizations nurture the continuous conversion of the two types of knowledge while producing new knowledge through socialization, externalization, combination, and internalization of knowledge (SECI model) [50]. One of the primary objectives of an organization's leadership is managing human and structural capital to develop renewal capital [51]. KM processes refer to knowledge identification, acquisition, creation, capturing, sharing, utilization, and transferring. Some of the KM processes are related to internal knowledge, and some others to external knowledge. KM practices consist of those knowledge-related managerial activities, systematic and consciously developed and implemented that aim to build competitive advantage and enhance firm performance [52].

Bratianu and Bejinaru [53] have stressed the value of knowledge as a spectrum of rational, emotional, and spiritual knowledge, as it is essential for grounded business decisions [54]. Knowledge is the cornerstone of a company's dynamic capabilities and strategies to create the necessary conditions for achieving competitive advantage [55]. Knowledge processes require integration, and companies implement them through leadership, management, technology, and organizational culture. Organizational knowledge comprises information and know-how transferred from individuals and social groups [56]. Knowledge can be leveraged from internal sources represented by human, structural, and relational capital or acquired or transferred from external ones, e.g., through partnerships with external stakeholders. Knowledge allows companies to build new capacities and sustain growth. Furthermore, firms utilize this knowledge with internal and external provenance to expand their competitive capabilities and take advantage of market opportunities.

Figure 1 presents a synthetic mind map of knowledge management, considering the main dimensions of the concept, which have also been included in the present study.

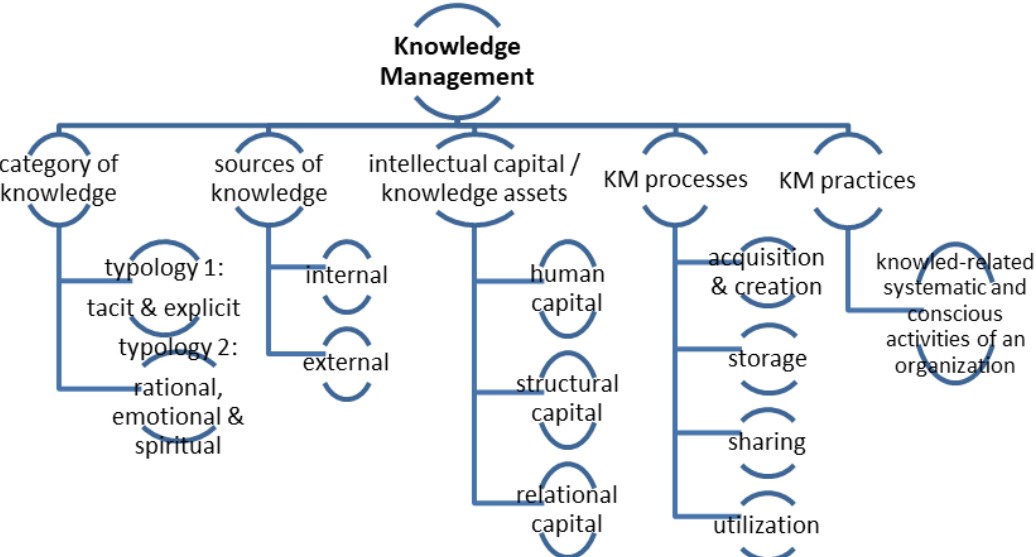

**Figure 1.** KM mind map.

KIBS could take advantage of a wide range of benefits related to effective and consistent knowledge management. Knowledge could generate the most important competitive resources for KIBS [57,58]; therefore, its proper management becomes strategic. The benefits generated to KIBS companies by knowledge management are very diverse and related both to their internal and external environments. Various studies have identified the fol-

lowing aspects generated by proper KM: better decision-making, enhanced availability of resources, expanded company knowledge, less or no knowledge loss, developed strategic analysis, development of a learning organization, increased agility, increased efficiency, increased competitiveness, increased productivity, increased teamwork and cohesion, a culture of innovation, stimulating up-skilling and re-skilling of the workforce, better collaboration and communication both within the company and with clients, increased customer satisfaction, development of external relationships and networks [59–63]. All these benefits are activated by KM, in relation to the perception of knowledge as a field manifested in three fundamental forms: rational knowledge, emotional knowledge, and spiritual knowledge [53,64].

*2.2. KM in KIBS and the COVID-19 Pandemic*

The COVID-19 pandemic is a complex crisis, which began as a crisis in the health system, and has subsequently extended to other connected systems, i.e., the economy, finance, education, etc. For managers, finding adequate solutions to deal with the crisis implies identifying and covering the appropriate knowledge gaps. Managing knowledge assets requires suitable knowledge competencies and capabilities. Furthermore, while learning during the crisis and incorporating the new knowledge, organizations must develop emergent knowledge strategies to cope with the crisis and recover [65]. In the framework of the pandemic, businesses have learned that the flow of knowledge management goes from relevant information gathering and developing agile repositories, to developing proper KM digital platforms, to reliable communication and networking, leading to organizational learning and the development of a culture of knowledge [66]. Therefore, integrating knowledge management with organizational processes is not only recommendable but also vital for agile development.

While deliberate knowledge strategies rely primarily on knowledge exploitation, emergent strategies are based on knowledge exploration, and they reflect the urgency brought by the crisis and its complexity. The institutional or strategic misalignment between a company and its environment can result in business failure if firms cannot adjust their behavior and processes or resources and capabilities, respectively, to comply and cope with the crisis management measures imposed by the COVID-19 pandemic [67,68].

According to Miles et al. [69], with some exceptions (sector and location dependent), KIBS proved to be resilient to the 2007–2008 financial crisis, recovering fast after a temporary hiatus. By comparison, "the Coronacrisis" has had a deeper impact on KIBS, especially in certain sectors like tourism, hospitality, advertising, and entertainment, due to the pandemic itself, policy responses, and socio-economic aftermath. However, KIBS in the health and IT sectors, for example, have generally benefitted from the demand for services to mitigate the effects of the crisis. Despite the organizations' lack of preparation and adequate mitigation strategies, the authors refuted the fact that the COVID-19 crisis is a "black swan" event since its likelihood has been long anticipated by public health bodies. Likewise, this crisis pointed to the need to enhance preparedness for future crises, like those related to envisaged climate change impacts. Before the COVID-19 pandemic, KIBS were usually involved in face-to-face exchanges with their clients, as tacit knowledge sharing was facilitated this way, but the COVID-19 pandemic has affected such interactions, which switched to online videoconferencing and webinars. This change has also impacted informal trust-building exercises through social interaction [69].

Only a few studies up to the present investigated the impact of the COVID-19 pandemic on KIBS [2,6,69–72]. One of the studies, related to the KIBS sector in Romania, documented maintenance of the performance level, in many cases, from before the pandemic due to the quality of the workforce and their familiarity with the technological shifts that became the norm during the lockdown [72]. The decrease in performance in some KIBS had been associated with disruptions in teamwork and knowledge sharing, especially in the first part of the pandemic. A positive aspect has also been documented, related to a perceived increase in the creativity of the staff trying to better cope with the new

challenges at work [72]. Organizational culture was one of the main factors that helped KIBS to adapt to the new challenges [70]. Shared values and open communication were facilitators of adaptation, helping overcome the difficulties such as the blurred boundaries between professional and personal lives, integration of new employees, coordination inside the team, etc. [70].

Since the COVID-19 pandemic has affected all companies, KIBS have had an important role in supporting their clients' resilience-building, compliance observance, and recovery efforts. On the other hand, KIBS had to confront their pandemic-inflicted challenges. From a business strategy perspective, KIBS had to adjust to the crisis context, by reducing expenses, accelerating digitalization, increasing IT security, enhancing online communication, and switching to remote working models. Some KIBS have developed solutions for supporting their employees' health and well-being [2]. Notwithstanding that many KIBS have launched information products dedicated to the COVID-19 pandemic (news, blogs, real-time data tracking, analysis reports, data visualization, etc.), either on their websites or on social media, to support the crisis management efforts by supplying accurate information and relevant advice.

Kirchner et al. [73] have empirically studied the managerial challenges associated with working from home during the COVID-19 pandemic vs. the employees' perceptions in Danish companies. Both categories seem confident in what concerns preparedness for this style of work. Still, managers felt their work tasks were more cumbersome and implied longer time allocated for various tasks since they could not avoid the extra hours spent in online meetings. Additionally, despite long work hours, the respondents stressed that work was less productive, as certain aspects would have normally been discussed informally in the office, thus avoiding formal meetings. The researchers' study pinpointed several main challenges for distance management: work (re) organizing, extra time dedicated to crisis management, more difficult communication and online collaboration with the employees, and a blurred perception of the employees' mental and emotional state. Kirchner et al. [73] concluded that distance management requires organizational capability learning and building, as it is not solely the managers' responsibility.

Like most economic organizations, KIBS have been greatly impacted by the COVID-19 pandemic, on many levels. A study by Abualqumboz [6], based on qualitative research on technology-based KIBS in the UK, shows that these companies went through three phases to accommodate the challenges determined by the pandemic: disharmony, normalization, and harmony. The first one consisted of intensive knowledge acquisition and adaptation to a new way of knowledge sharing. The second phase consisted of the stabilization of processes related to knowledge sharing and co-creation with clients. Still, overwork and improvisation were quite often considered during this phase. The last one, harmony, was characterized by compliance with the new context, and investment in reaching resilience.

One of the aspects that differentiate KIBS from other organizations is they can help their clients to overcome the difficulties associated with the pandemic [56], especially since KIBS are credited more than other firms with networking and co-creation with their clients [1]. Some KIBS even benefited from increased demand for their highly specialized services and products, needed even more during the pandemic. As such, KIBS did not only have to deal with significant changes in their organizations, just like every other economic actor, but they also dealt with their clients' struggles.

In a context of unpredictability and crisis, existing information and knowledge are no longer able to fuel predictions about the immediate, medium, or even long-term future or ensure a sense of control. In this environment, the increased adaptability capabilities that KIBS own allowed them to become universal problem solvers [73]. Analyzing response strategies developed for internal and external (client) challenges in the COVID-19 environment, one concludes that KIBS organizations represent potential key players in the context of future crises and uncertainty [2]. As the distinctive mission that the KIBS assumed during the COVID-19 pandemic facilitated the navigation of troubled waters

for both themselves and their clients, they likely enforced their strategic position on the business market for future similar situations.

## 3. Materials and Methods

Having in mind the uncovered evolution of KIBS knowledge management practices during the pandemic, the present paper focuses on analyzing the specific challenges determined by the COVID-19 pandemic in relation to knowledge management in KIBS in the specific situation of Romania. A sample of four successful Romanian KIBS was investigated, as they are representative of the three main categories of KIBS (t/p/c KIBS).

The study is based on qualitative analysis since it aims to understand the dynamics of real-time transformation related to the COVID-19 pandemic. This would allow for a nuanced and more extensive, personalized analysis. To ensure the reliability of the investigation, we used an interview guide based on semi-structured open questions [74]. To ensure a reliable multiple case study investigation, we had in mind two sampling criteria: the domain of the companies to be relevant for the investigation to reflect several types of KIBS; the companies to be well-known and have a solid reputation on the market [75]. Convenience was also a factor that determined the content of the final sample.

The interviews aimed at two main lines of investigation related to the COVID-19 pandemic. To answer the research questions regarding the challenges for knowledge management in KIBS caused by the COVID-19 pandemic and how this crisis has influenced knowledge management performance, several steps were taken.

First, when exploring the new challenges for knowledge management generated by the COVID-19 pandemic, the revealed areas referred to mobility challenges in the context of social distancing, knowledge systems security challenges as cyber attacks intensified all over the globe, and even technical challenges. For example, during the pandemic, a new layer was added to KIBS in creative industries activities represented by communicating brands' responsibility and readiness to serve their clients in the new context. Next, focusing on knowledge performance has also been impacted as new knowledge was required for KIBS to provide relevant solutions to their clients. As such, KIBS were among the first to acquire, integrate, and use new knowledge in their practices.

The interview guide included many lines of investigation, including knowledge management in the company; knowledge acquisition; knowledge storage; knowledge sharing and protection; knowledge valorization; and KM performance. Data were collected during May and June 2022. The interviews were conducted online, using various video platforms to make respondents more comfortable. Most interviews had a duration of around one hour. They have been recorded with the participants' informed consent. The acquired content has been transcribed verbatim and processed by hand. Further on, the data has been analyzed following a theme classification method, which included aspects related to the organization's KM strategy, its KM processes, and KM performance. This research focuses on the impact of the COVID-19 pandemic on the organizational KM processes and overall KM performance.

The KIBS sample consists of 16 interviews with top managers from four organizations, technology-based and professional companies, one of which includes creative services. Company K, whose primary services include digital marketing and PR, has introduced more recently additional services such as photography or graphic and web design. The interviewees are presented in Table 1.

All investigated companies have long experience in the Romanian market, are local firms, and are well-respected organizations. For example, Company P has had a presence for 16 years as an advertising agency, and is known as a knowledge incubator for young passionate professionals in the field.

E is a marketing consultancy company operating since 2008 with 41 employees until the end of 2021. The company specializes in strategy and digital commerce engineering, and the company's marketing consultants offer a range of services, which include helping to devise, plan, and implement digital marketing campaigns across channels; training

other marketers on best practices and technologies; evaluating current marketing efforts and making suggestions for improvements; and offering solutions for workflows or new methods for reaching and converting consumers. The company serves the European markets. The industry orientation is represented by e-commerce, multichannel engagement, social media, and digital disruption.

**Table 1.** Structure of the sample.

| Interviewees | Job Position, Year of Experience in the Company | Company | Domain |
|---|---|---|---|
| E.H. | CEO, 14 years | E | Marketing consultancy |
| A.H. | Human Resources Manager, 6 years | E | Marketing consultancy |
| C.D. | PM leader, 4 years | E | Marketing consultancy |
| M.P. | Performance coach, 1 year | E | Marketing consultancy |
| C.B. | General Manager and co-founder, 10 years | K | Digital marketing & PR agency |
| V.D. | Executive Manager and co-founder, 10 years | K | Digital marketing & PR agency |
| R.F. | Social Media Manager, >4 years | K | Digital marketing & PR agency |
| C.S. | COO, >4 years | K | Digital marketing & PR agency |
| D.N. | Managing director, 16 years | P | Advertising agency |
| M.T. | New business director, less than 1 year (2 months at the moment of the interview) | P | Advertising agency |
| L.I. | Creative director, 5 years | P | Advertising agency |
| R.D. | Strategic planner, 4 years | P | Advertising agency |
| L.S. | CEO, >5 years | M | Technological company |
| M.H. | co-COO, >3 years | M | Technological company |
| A.B. | co-COO, >6 years | M | Technological company |
| M.I. | CFO > 7 years | M | Technological company |

K has been on the market since 2012. It has four associates and 35 employees. The company offers specialized services in the areas of social media account management, online ads campaigns, SEO optimization, PR and influencer marketing, and graphic and web design. It also organizes corporate workshops and training courses. K's website stresses the agency's focus on the value of continuous learning and cultural openness. It is considered an actor that stimulates professional development in the field.

P is a company offering advertising services as of 2006, with a little over 30 employees at the end of 2021, and which provides services such as integrated marketing campaigns, events management, shopper experience, digital marketing, social media, public relations, internal communication, and others. While the structure of the services is quite strict and the process flows are standardized, the final outputs are tailored to each client's needs, the field of activity, and the target group of consumers. The company is led by the managing director, who is also the founder, together with the board of management, formed by the respondents to our survey.

M is a software-engineering company that was founded in 2015 by a Romanian entrepreneur and has only one associate, the founder being the owner of the firm. It is said to provide excellent IT services, covering the entire software development circle: from defining the idea to implementation and maintenance, and support. The company is industry agnostic and has experience and expertise in various fields, such as healthcare,

retail, fintech, food tech, pharma, etc. The organization employs the latest technologies to deliver high-quality services to its partners, most of them being international. The company has a mostly flat organizational culture, with departments working with each other, and the power of the team being the scaling instrument. With 40% growth year to year, the firm is considered one of the fastest-growing IT companies in Romania, and it currently has over 250 employees.

## 4. Results

### 4.1. Knowledge Management in Romanian KIBS

The interviews revealed many similarities between the investigated cases, but also some specific approaches and understanding of the KM practice. In most cases, KM is not formalized overall, but there is a high interest in making it more formal and effective, especially in relation to knowledge sharing, while the organizational cultures encourage knowledge sharing and networking both inside and outside the organizations. Digital tools are the preferred instruments when considering both knowledge sharing and knowledge management in general.

In the case of Company E, the interviews revealed an informal approach in terms of an organizational culture that encourages open communication and collaboration. One observes that there is untapped knowledge within its workforce, which is lying dormant.

The interviewees for Company K stressed the work process flexibility and open organizational culture encouraging an informal exchange of ideas and practices. The company did not have a formal KM strategy or procedures, and some KM performance assessments have been temporarily discontinued due to the pandemic. A relatively wide variety of KM approaches has been identified, consisting of codification of more complex knowledge, permanent access to learning opportunities, transfer of knowledge with external stakeholders, networking, and collaboration with partners. The managers interviewed consider that staff needs not only technical competencies but also soft skills, communication, and relational skills

In the case of Company P, some formal KM processes have been identified, such as performance assessment, structured data storage, onboarding manuals, and internally designed and conducted research. Also, some semi-formal practices were spotted, consisting of structured internal and external knowledge-sharing activities, providing access to L&D resources, and promoting a stellar knowledge-sharing culture. KM is also present at the highest strategic level since the strategic business priorities include knowledge gap identification, acquisition, and application dimensions; and knowledge sharing is encouraged across organizations and with stakeholders.

For Company M, the interviews revealed a non-hierarchical culture in the context of a flat organization with some strong holacracy principles. It has approached some steps toward a structured approach to doing KM. The process is hindered by the desire to find a balance between a well-defined strategy and the flexibility needed to ensure the freedom of initiative. At the organizational level, a transformation of rational knowledge into emotional and spiritual knowledge has been observed.

Table 2 centralizes the main aspects identified concerning the KM processes in the interviewed companies. This table synthesizes the similarities among the companies concerning the investigated processes. It shows some flexibility, references set both inside and outside organizations, as well as the relevance of leadership.

As indicated during the interviews, in many cases KM is the main responsibility of senior managers, though the staff is also involved. KIBS companies acquire knowledge from a variety of sources, internal and external, including from the competition, and online resources are favored. Collaborative digital tools are used for knowledge documentation and storage. All investigated KIBS appear to nurture an organizational culture that stimulates knowledge sharing not only among internal stakeholders but also with external ones. Most KIBS employ knowledge management practices that ensure fast access to relevant resources. While some companies adopted a limited codification strategy (people-to-documents) for

managing knowledge, others chose the personalization strategy (person-to-person), which is more frequent for small companies offering customized services [76].

Some general challenges have been identified. In the case of KM identification and acquisition, various knowledge gaps have been considered, but also access to knowledge in the context of constant change. C.S. best described the meaning of knowledge inside Company K: "*Knowledge includes specialized know-how concerning digital marketing (channels, campaigns, press releases, etc.), but also all the information we have about our sector, as well as the networking know-how*". C.B. stressed the difficulties related to updating knowledge in digital marketing, as the technological pace is fast and knowledge becomes quickly obsolete. When new competencies and skills are required for the job, the managers, who continuously monitor the developments in the market, organize access to customized, hands-on training. Apart from that, employees have permanent access to online training, learning platforms, etc.

**Table 2.** The main lines followed by the KM processes in the Romanian KIBS.

| Knowledge Acquisition and Creation | Knowledge Storage | Knowledge Sharing | Knowledge Utilization |
|---|---|---|---|
| External resources are very diverse Online resources are the norm Competitors are considered sources of critical knowledge The persons responsible for new knowledge identification are generally in senior positions | The main instruments used for documentation and storage are collaborative applications based on the cloud, and databases, but also simple tools such as emails, WhatsApp groups The leader of the project tends to be in charge of the storage of knowledge | Most respondents related to organizational cultures that encourage knowledge sharing not only inside the organizations but also with partners and clients The approaches adopted to stimulate and ensure knowledge sharing are multiple: meetings, internal programs, coaching sessions, etc. | Flexible approaches have been identified, aiming to use relevant knowledge promptly People and technology are the channels for knowledge application Client cooperation is desired and activated |

In the case of Company E, most challenges have been associated with identifying knowledge gaps: "We are facing challenges with soft skills training that did not solve the gap because what is taught in the respective course is not always implemented. We thought of having a post-training discussion at the end of the year to evaluate the training. Therefore, some training did not leave its mark on the knowledge of my colleagues." (E.H.), "We are facing challenges with time management; the company is addressing the challenges with determination and support." (A.H.), "The moment we identify knowledge gaps, we ask ourselves about the way the employee was trained because we cannot hide incompetence for a very long time. We identified within the company, employees who do not have communication skills, and we decided that those employees should not meet with clients before we train them to improve their communication skills." (M.P.), "I had colleagues who although according to their CV experience and employment-related competencies, failed to meet our expectations. We are addressing the mentioned challenges by coaching the team member and assigning a mentor to him to improve his knowledge and skills. The consequences of not addressing the mentioned challenges were a greater need for resources to be allocated to a certain project and destabilize other projects." (C.D.)

KM sharing has been also related to various challenges. The respondents insisted more on these aspects, indicating the high level of importance of this process. Lack of time, limited access to information, and even the skills of some leaders have been mentioned.

Several aspects related to new members of the team and their limited technical skills have also been considered.

In the case of KM utilization, key elements mentioned were, again, the knowledge gaps and close cooperation inside companies. For instance, in terms of knowledge application, Company K underlined that, in the field of digital marketing, the necessary knowledge changes continuously. Therefore, knowledge gaps are rather frequently identified, either by the employees themselves or the supervisors. Newer employees receive the help of more experienced staff when struggling to apply new knowledge and benefit from customized training. Staff is given the freedom to experiment and come up with new ideas and solutions to the client's problems. Another challenge identified was the difficulty to integrate available knowledge due to the limited abilities of the staff related to analyzing and understanding knowledge, especially that existing or acquired from outside the company.

*4.2. Challenges and Solutions for Effective Knowledge Management in Romanian KIBS during the COVID-19 Pandemic*

4.2.1. New Challenges of KM Processes

In addition to the challenges specified in the previous section, the pandemic introduced new challenges that are presented in this sub-section. Most of the additional challenges are related to the processes of knowledge transfer.

KM Creation and Acquisition

A main challenge specified especially in the case of the companies that benefited from the opportunities associated with the pandemic and developed their activities was the difficult control and handling of knowledge in the context of accelerated growth.

KM Storage

Challenges associated with KM storage are related to designing procedures, which have been made more difficult by remote work. For instance, R.F. and C.S. of Company K pointed to the use of knowledge codification procedures within the company. The pandemic made informal communication that facilitates knowledge storage (as well as knowledge sharing) more difficult and time-consuming.

KM Sharing

Respondents mentioned increased difficulties related to this process. Company P representatives, for instance, indicated unanimously the knowledge gaps challenge that seemed to have deepened during the COVID-19 pandemic. On one hand, D.N. and M.T. mentioned the skills gap, while, on the other hand, R.D. provided meaningful details about industry knowledge gaps, as "( . . . ) clients cannot provide information that we need. In these cases we research online, we reach out to the industry studies, and we perform studies using internal resources (observations, surveys, in-depth interviews, online questionnaires). There were consequences when the client complains about the quality of the product, after not providing the requested information. There is always a misalignment between client knowledge about their industry (based on selling reports, client support reports, etc.) and our knowledge about their industry, which needs to be constantly mitigated and this represents a challenge".

To share knowledge inside the organization and sometimes to transfer it to external stakeholders, company K mainly employs the tools of its trade, social media, and social networks, such as Facebook, Instagram, LinkedIn, TikTok, Twitter, etc. R.F. underlined that "knowledge is delivered in the final outputs towards the clients". K's organizational culture encouraged face-to-face interactions, informal (tacit) knowledge sharing and hands-on assistance to newcomers or less experienced employees. From this perspective, the COVID-19 pandemic was a challenge for the company's practices. Likewise, since Company K used to hold regular meetings—including face-to-face—with clients to keep track

of their business development, provide them with updated feedback, and adapt the digital marketing strategies, the pandemic affected the physical interactions, too.

Remote work was a significant challenge for all companies investigated. According to the qualitative research findings, the COVID-19 pandemic impacted Company P mostly from the perspective of the location of work. Due to the national lockdown, employees were forced to work remotely. More specifically, this generated unprecedented pressure on the digital system, which has been referenced by respondents as a crisis that had to be solved.

With regards to Company M, the COVID-19 impact had a strong influence on the way the knowledge-sharing process was conducted: being a relatively small company in 2020, the people within the organization used to share lessons, information, and general knowledge in an informal way, face to face, while working in the same office. Switching to a 100% remote context, the organization faced challenges in creating a new context for people to share their knowledge, without diluting their flat organizational culture.

Based on the data gathered and assessed during the interviews, it can be noted that COVID-19 played an important role in the evolution of Company M, being a challenge that helped them think outside of the box and come up with innovative and novel ideas to secure the organizational culture and knowledge management processes. Even though the organization encouraged working remotely before the pandemic, after March 2020 things changed quickly as all organizations went remote: "Working from home and remote work make it a bit harder, because in the past the knowledge was shared naturally in the office, but now we need to think about other ways of doing it" (A.B). The need to react swiftly and produce new ways of helping people share their knowledge and acquire new ones, in a context that lacks face-to-face interaction, has been an essential aspect in the past years. As L.S. mentioned, "Remote work, correlated with our accelerated growth, made it harder for us. We did not build a system to support knowledge sharing during remote work to flow naturally, but we learned that having a buddy system in place, dedicated sessions to discuss the projects, and transparently communicating information helps. We are still learning and doing our best".

KM Utilization

Companies apply knowledge by transforming it into products and services for their clients. Sometimes gaps between the company's and client's knowledge are identified, and the pandemics increased these gaps as well as made the communication needed to mediate these aspects more difficult.

A main formal impact on the inter-company KM covers the integration of new products developed for their clients, backed by new knowledge acquisition, application, storage, and protection. This process has been hindered by the pandemic, with additional efforts being necessary. For instance, in the case of Company P, from its website, we notice that the firm developed a specialized segment of projects covering COVID-19 effects in the form of marketing campaigns integrating references to social distancing and mask-wearing. The key role of this new products division is to build clients' brand awareness around social distancing and health safety within their industry-specific activities and communicate the brands' success in adaptation to the new reality.

4.2.2. KM Practices

The data acquired during the interviews revealed that Company K does not embrace formal strategies and processes, as it sees it fit for a small, flat-hierarchy firm to maintain flexibility to the largest extent possible. Notwithstanding, K does have knowledge management processes and practices in place, which appear to be implicit for KIBS in general. The firm employs digital tools and informal exchanges for knowledge acquisition, storing, and sharing. The company's practices, which nurtured tacit knowledge sharing through informal meetings and face-to-face interactions, have been challenged by the switch to virtual meetings. Adapting to remote work during the COVID-19 pandemic brought not

only disadvantages as those mentioned above but also positive changes. Due to the increased demand for digital services during the pandemic, the firm had to employ new staff, and thus, the company went through accelerated growth. The hiring process brought difficulties from the point of view of assessing the candidates' technical and soft skills, which normally happened in face-to-face encounters. The online working processes also proved difficult from a psychological perspective, and the employees "*seemed tired of the solely online learning opportunities*" (R.F.). Furthermore, given the challenges mentioned previously, the evaluation of the employee's performance in terms of knowledge acquisition and application, which were regular, had to be discontinued since the beginning of the pandemic.

Company P faced internet security infrastructure issues during the pandemic period, as the virtual private network (VPN) could not support at times all the traffic caused by employees working from home. At the same time, this represented an opportunity to grow, as indicated by R.D.: "( . . . ), because in the pandemic environment we identified some issues with the VPN and we fixed it".

Another challenge that Company P faced is represented by the advancement of social media in the past couple of years and the difficulty of keeping up with technological requirements of performance in the field. When asked if their company is facing any challenges, D.N. mentioned: "Yes, I could say specifically to social media, where things are evolving at a very high speed, and there are a lot of technical things that you need to know, while you might be very good at certain components (as content or community management) but you don't master ads' area".

In the case of Company E, when asked about the general challenges they have encountered concerning KM within the company, the interviewees pointed to several aspects, though, in general, their perception has been that there were few difficulties, and consequences have never been major: "We are currently thinking of a better option to address these changes differently and to introduce an internal platform where we can address the challenges." (E.H.).

Challenges related to the workforce also have had to be considered during the pandemic: "Due to the increased volume of deliveries, we are looking for new colleagues with required skills that we are looking for, but after investing time in training them, there are colleagues who gave up" (E.H.), "resistance to change" (A.H.&M.P.), and "finding qualified personnel on certain technologies with certain knowledge and skills to join us." (C.D.) Company E developed a staff motivation project to better cope with the challenges associated with the pandemic. The pandemic also permitted another evolution related to the workforce—because of the new working-from-home approach, team members from various cities have been hired. This opened new possibilities and "we want to do much more than we are doing at the moment." (E.H.) Also, to better cope with the challenges of the pandemic, a staff motivation program has been developed.

Another challenge that Company M observed, mostly starting with 2022, but as a result of the global pandemic, was the global openness of the IT industry, translating into the possibility to work remotely for almost any company in the world. This resulted in an even more competitive market to bring in new talent and ensure a constant flow of knowledge management, not only to fill in the gaps but also to facilitate the acquisition of new and relevant knowledge needed for innovation: "Recruit new people in a very competitive market and encourage the old ones in the organization to share their experience and knowledge to teach others is an interesting challenge we have been facing in the past year"(M.H.). The market is more and more competitive, and it became essential to create new approaches, from the ways of performing the technical work to improving and innovating organizational culture and people strategy, which are deeply connected to KM strategy.

Lastly, the most prominent challenge Company M has been facing is generated by the accelerated growth they register every year. When correlated with the COVID-19 global situation, the company found itself in the middle of handling new remote processes and

strategies for more and more people. Growing by 40% each year implies a constant change at the organizational level, as there are different points of company maturity. Looking back, all interviewees pointed out the way the KM practices changed over the years, especially after 2020 and COVID-19: "Yes, they changed several times, as we had various points of transformation of the organization, from 50 to 100, we had to reinvent all the processes, the same way we are doing, now being more than 200 people" (A.B). Also, both COVID-19 and the fast growth pushed the company towards a more structured approach in KM practice, which is essential for the organization's evolution and overall strategy, which was mentioned by CEO L.S.

Table 3 synthesizes the main challenges and corresponding solutions, associated with the COVID-19 pandemic. It appears that the investigated KIBS took measures to adapt to the new reality, opting for tactical solutions, that could offer fast remedies and quick wins.

**Table 3.** KM challenges and solutions.

| Intra-Company KM | |
|---|---|
| **Challenge** | **Solution(s)** |
| Work remotely | Infrastructure development<br>Coaching sessions |
| Facilitate cooperation remotely and knowledge sharing | Enhanced online discussion opportunities |
| Increased access to knowledge and knowledge acquisition | Infrastructure development<br>Employee-oriented programs to support them psychologically as well as motivate them |
| Internet security | Infrastructure development |
| Staff-related challenges (knowledge gaps, cooperation, pressure, etc.) | Staff-motivation program<br>Training programs for new employees |
| **Inter-company KM** | |
| **Challenge** | **Solution(s)** |
| (New) knowledge sharing with clients | New products<br>New division dedicated to COVID-19-related communication |
| Company-client knowledge gaps | Research on available media<br>Increased communication |
| Knowledge application | Tighter cooperation with clients |
| Maintain reputation and trust among clients | COVID-19-related information/marketing communication campaign |

### 4.3. KM Performance and Customer Relationship during the Pandemic

KM has been explicitly indicated as the "basis for performance", and respondents seemed satisfied with the associated outcomes. Nevertheless, generally, the investigated companies do not measure KM performance explicitly, in a formal way. Additionally, some aspects that have been previously evaluated might have been put on hold because of the disruption of activities and processes related to the COVID-19 crisis. During the pandemic, additional attention has been given to digital performance. For instance, Company P developed a digital performance evaluation map.

Another important outcome associated with KM is effective customer relationships, as C.B. from Company K explains: "I believe the relational knowledge concerning customers is the most important [outcome of KM] in our company." Knowledge is explicitly associated with better customer services and relationships by respondents in all companies. For instance, E.H., the CEO of Company E, states: "The value of knowledge is very high because with its help we manage to serve our customers better. The knowledge in our company

means knowledge of different industries, digital marketing services, and knowledge of platform development technologies".

Knowledge sharing has been identified as of high relevance for the KIBS interviewed. Therefore, also diverse challenges have been discussed in relation to customers and various stakeholders: consistency, selecting knowledge shared, feedback acquisition, and expanding knowledge of customers. The aim is to ensure better services as well as enhanced customer relationships.

The data gathered during the interviews revealed that the restrictions of social distancing from governments on customer movement have created mostly obstacles for a company to connect with consumers. Therefore, the main medium considered by KIBS (and not only) for connecting both company and consumers has been via online and digital marketing. The pandemic forced companies to adopt working from home, which might not be suited to everyone's personality or ability and create difficulties in monitoring employee performance. Also, upgrading skills and staff development was challenging when the staff was not in close physical proximity. This evolution initially determined some disruptions within the knowledge flows between the organization and its clients, as well as outside stakeholders. For instance, in the case of Company P, R.D. explained that the most significant challenge is obtaining industry-specific knowledge, especially from the clients. Feedback from clients is generally considered very important, including for building and implementing other projects. Therefore, the disruption initially introduced by the pandemic was a significant challenge to overcome.

A more general challenge of fast and effective communication with consumers was initially enhanced by the pandemic: "Since we are connected to the external and international environment, we must be aware of new appearances and practices; all clients want faster, better and more and we have to face these challenges"(M.P., Company E). This comes also over difficult open cooperation with some clients, who "are not interested in regular meetings, which can be frustrating, since we are aware of the importance of such exchanges" (C.B., Company K). Communicating online became the norm during the pandemic; therefore, part of these communication problems have been overcome via online video conferencing.

The approach to KM performance is not strategic in nature, in the investigated KIBS. It is rather tactical, and, therefore, reactions are connected to the dynamics of the challenges. The main aspect mentioned is connected to customer relationships, which would lead to better customer service.

A second dimension specified was connected to employees' handling of knowledge, including working remotely, using the internet as a communication tool and even working channel, extending the workforce and work responsibilities, determining new challenges and fast adaptation, and influencing the performance of KM practices. The solutions to increase KM performance, as presented in Table 3, consist of infrastructure development, increased cooperation and better knowledge flows, new hiring procedures, and taking novel approaches to train new employees.

## 5. Discussion

Although KM is widely considered the foundation for performance, most of the Romanian KIBS do not have a comprehensive KM formal approach, although some aspects are assumed, and the importance of its adoption is widely recognized. A similar situation has been identified by other studies, developed before the pandemic, in other markets, which observed a wide interest and organic developments in assuming KM strategies rather than strategic planning [57,58,77]. KM management is under a tactical approach, even if it is considered strategic in nature by the KIBS representatives.

Measuring KM performance seems not to be considered very important, also on a background of diffuse satisfaction with the results of the knowledge management and sharing processes across the investigated organizations. Therefore, the positive impact claimed for KM in relationship with the company's performance might be biased and not

supported by reality. Practices should be changed concerning this topic. This situation denotes the need for more professionalism in approaching and formalizing knowledge management in Romanian KIBS, considering many aspects from design to implementation.

Relational capital proved to be a main concern for KIBS in Romania. KM performance is tightly connected to effective customer relationships, which are linked to the success of customer service. Therefore, the study suggests that respondents recognize that customer relationships mediate the positive impact of KM on organizational success. The means to stimulate this mechanism are rather tactical in nature.

A more strategic approach to KM might have helped better deal with the new challenges: disruptions inside the teamwork that paradoxically emerged during exclusively online cooperation, similar disruption related to customer relationships, valorizing new opportunities, facing more unknown factors when fast hiring new employees, widening knowledge-gaps both considering internal and external environments, etc. Not only the pandemic-related new challenges led to disruptions in KM formal and informal activities, but also the increased activities in certain compartments determined the temporary suspensation of some KM practices and evaluation procedures. On-the-go adaptation leads generally to accelerated adoption of effective practices, but sometimes the quality of the assumed solutions might not be the most adequate/strategic one. Therefore, an audit of these changes made under the pressure of pandemic-related challenges and fast business growth is necessary.

One important and widespread aspect discussed by most interviewees is connected to an increased number of knowledge gaps identified both at intra- and inter-company levels, in association with the pandemic. Three main types of knowledge gaps were mainly considered: related to KIBS' general management processes, employees/work teams, and clients/customer relations. Some of these knowledge gaps are not new for organizations but enhanced in the first part of the pandemic. Normalization to the previous state has been registered. Still, knowledge gaps represent current challenges that might be addressed by effective KM, since knowledge is the most important competitive asset of KIBS with specific characteristics due to its intangible characteristics and enhanced dynamics. Also, research shows that proper KM help KIBS be more resilient during the pandemic [71]. Therefore, KM should not be optional for KIBS but at the core of business strategies.

COVID-19 was an opportunity for organizational development for the investigated KIBS. This was not the case for all KIBS; nevertheless, the demand for some services or products provided by KIBS also decreased [69]. Some companies increased significantly in terms of employees, but also business outcomes. This led to two main challenges: hiring new and adequate specialized employees, and also training new staff during the pandemic; ensuring new and relevant services and products satisfy the shifting needs of customers. Also, this necessitated widening digital infrastructure and processes, both considering internal processes and customer-related ones. The new online cooperation between team members, as well as the online hiring of persons, proved to be challenging since gaps have been reported between the evaluation at the hiring interview and the actual competencies and skills of the new employees. The pandemic showed that evaluation of and operationalizing exclusively online rational knowledge is not so straightforward. Also, the pandemic brought additional challenges concerning emotional and spiritual knowledge, and additional accommodation programs were helpful.

For KM adoption, the pandemic was not a missed opportunity. Before the pandemic, despite a wide understanding of the practical and strategic relevance of KM, only some organizations had implemented formal associated approaches or evaluation procedures. During the pandemic, companies had to adapt, and adopting more formal KM practices was part of the development approach. At the same time, some formal processes related to KM have been disrupted; these are mainly connected with workflows, direct communication, evaluation, and control. Nevertheless, the adapted digital infrastructure offered adequate solutions.

Only partial knowledge management is incorporated into business strategy. Some organizations perceive knowledge transfer, both inside and outside the organizations as highly strategic. In this framework, the negative impact that the pandemic had on customer relationships, as well as inside teams, in the beginning, setting barriers and delaying processes, was an important challenge that was solved both formally and informally.

## 6. Conclusions

KM in KIBS is an extremely relevant topic, but academic research in the field, including in the main academic flows, is rather narrow [8,24–28,30,57,58,78–86]. Also, many authors ignore the main attributes of knowledge, which are intangibility and nonlinearity [53,87]. Therefore, the present research adds to the existing thin field of investigation. The existing studies concentrate on how KM can be managed for innovation in a collaborative framework with industry and higher education, as well as on the approaches assumed formally or informally by KIBS in the field of KM, with a special focus on intellectual capital and knowledge sharing, both being main concerns both in the research and practice of KM.

The previous studies [70,72] on the pandemic's influence on the Romanian KIBS's KM practices and procedures stressed the work environment transformations and the importance of organizational culture. The present study enhanced these aspects, adding to the picture the important role and associated disruptions and solutions related to knowledge sharing, but also with various intra- and inter-organizational knowledge gaps.

As revealed by studies on other markets [6], the Romanian KIBS also went through three stages in their adaptation to the pandemic-induced challenges: disharmony, normalization, and harmony. The adaptation of KIBS to strategic shocks is a relatively linear process, which proves the high resilience and flexibility of these organizations, which mobilize a mix of strategic and tactical approaches to this end. The investigation permits a better understanding of the way KIBS organizations cope with crises from the perspective of knowledge management. The analyzed KIBS companies prefer mostly an informal approach to knowledge management, closely associated with the personalization strategy (person-to-person). The study shows that in their effort to cope and adapt to the new reality brought by the COVID-19 pandemic, KIBS have chosen practical, tactical approaches to solve the challenges of acquiring and creating, storing, sharing, and utilizing knowledge effectively, supported by digital technologies. Some of the practices established before the health crisis have been disrupted, while new ones have emerged. At the same time, learning from the lessons of the pandemic, KIBS will have to consider updating their KM processes and practices and building reliable knowledge bases.

The findings permit the understanding of how crises could be both disrupters and stimuli for KM strategies and effective practices. To some extent, crises push companies to adopt more strategic approaches at an accelerated pace but also determine them to solve some tactical issues to cope better with the direct pressure putting on hold some more strategic dimensions in some companies. Leadership is essential, as confirmed by the present study, especially in countries such as Romania, where the power distance and uncertainty avoidance are so high [88,89]. At the same time, the investigations allow us a more nuanced view of the general challenges related to KM in KIBS, as well as how to better design customer relationships in this context.

### 6.1. Research Limitations

Limitations of the research are primarily related to the number of organizations investigated. Having this in mind, we interviewed a significant number of managers from each organization, with diverse responsibilities, to understand better the many facets of the KM activities in those organizations and their diverse implications. Another aspect worth discussing is that the respondents are extremely active professionals, and the themes approached quite diverse and investigated many facets. Therefore, on one hand, the interviews could not take too much time, and, on the other hand, the risk of too brief explanations was high. This vulnerability was also diminished by the number of interviews

considered for each organization. Another limitation that was mitigated by the number of interviews and in-depth discussions is the biases of the interviewees towards their role and how effective their organizations were under pressure. Additional cross-checking and supplementary information have been considered from the online environment, such as companies' websites and communications. We also highlight that the companies selected are leaders in their fields of activity and, therefore, probably reflect the best practices in the field. Thus, the results are relevant and contribute to understanding the challenges and best solutions adopted so far by Romanian KIBS.

*6.2. Research Implications*

The study confirmed that managers are key actors in dealing with crises in Romanian KIBS, as well as the high resilience of these types of companies in the face of crises. The description of the main challenges and adopted functional solutions permit managers to reflect on the processes associated with the pandemic, to observe the role that knowledge and knowledge management played both at the intra-organizational level, and in the inter-companies context.

Since leaders are key players who can mitigate challenges and stimulate faster adaptation in times of crisis, they are expected to take charge and closely guide their teams. We recommend exploring how flatter organizational cultures might contribute to faster adaptation and stimulate more creativity and innovative approaches inside companies, as well as considering their relationships with the external environment. Also, recognizing at the managerial level the importance of formal and strategic KM is not enough without strong approaches along this line. Romanian KIBS companies should reconsider their priorities and implement such strategies, valorizing the experience gained during the pandemic, embedding risk management dimensions, and considering the increasingly turbulent environment that they have to cope with.

The study revealed the importance of the infrastructure in ensuring effective and timely knowledge management, both inside the organization, and in connection to clients and stakeholders. The tendency is to adopt the available, widespread solutions, which are functional and show a pragmatic view. Nevertheless, solid and more reliable technological infrastructure and personalized digital solutions might determine stronger resilience, faster adaptation, and prompter knowledge sharing and cooperation inside the company and with clients. Dynamic capability would enhance KIBS's performance.

The investigation proves useful also for KIBS' stakeholders. It documents the openness for cooperation, how KIBS values client and stakeholder relationships, and the desire for and even active involvement in professional development in the field. Therefore, stakeholders might accept this cooperation proposal and contribute to developing functional networks and enhancing professional cooperation in the Romanian market.

In the end, we mention the need for a more formal approach to the evaluation of the knowledge management processes, procedures, and strategies. All respondents appreciate the positive impact of KM on the company's performance, but no actual instrument for measuring this impact, or the effectiveness of KM in general on various components, is implemented. Companies should also concentrate their managerial strategic efforts on this direction.

**Author Contributions:** Conceptualization, A.Z.; methodology, A.Z. and project team; validation and formal analysis, all authors; investigation and data curation, E.D., A.-N.I., R.-M.S. and B.-R.S.; writing—original draft preparation, all authors; writing—review and editing, A.Z.; supervision, A.Z.; project administration, A.Z. All authors have read and agreed to the published version of the manuscript.

**Funding:** The authors greatly acknowledge the financial support from the European Union's Erasmus+ Program under Project Reference 2021-1-EE01-KA220-VET-000032944. The authors further greatly acknowledge the efforts of the other partners in this project as, otherwise, this paper would not have been possible.

**Institutional Review Board Statement:** This statement was not necessary in this institution.

**Informed Consent Statement:** Informed consent was obtained from all subjects involved in the study.

**Data Availability Statement:** The data presented in this study are available on request from the corresponding author.

**Conflicts of Interest:** The authors declare no conflict of interest.

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
