# Peer review of "Managing Knowledge in Romanian KIBS during the COVID-19 Pandemic"

_knowledge, doi:10.3390/knowledge3010002_

Round 1

Reviewer 1 Report

All in all, the paper and the idea behind it is very compelling. However, there are a few things that can be improved upon.

In the introduction/literature review - have you considered establishing a link between the research and the resouce-based view of strategy design (as described by Barney, Rumelt, Schmalensee, Grant, and others)? The prevailing idea of strategic management and value management in today's environment is that it should be based on resources, knowledge, skills and experience of analysts, planners and managers. This connection could broaden your research from a theoretical standpoint, since you are in effect investigating how KIBS are coping with the strategic shock (as described by Mintzberg) of the COVID-19 pandemic, and how these firms are reactively/proactively trying to deter risks and exploit opportunities in a highly complex and turbulent environment. All of this might help you create a more effective conclusion.

The research design is something that should be explained in more detail, especially the hypotheses, which are missing. The discussion of the results is not coherent, which reflects in the conclusion.

Author Response

Dear reviewer,

Thank you for the invitation to revise and resubmit our manuscript entitled "Managing knowledge in KIBS during the COVID-19 pandemic" and for your kind and constructive feedback.

Please see below (in red) our responses to your concerns. We believe that all the recommendations helped us increase the relevance of the paper, and contributed to its improvement. We are grateful for all your significant and valuable suggestions.

We will upload two versions of the revised manuscript – one with track changes to easier follow the modifications and one without track changes for a clearer view of the present form of the research. We hope we could correctly grasp and incorporate your recommendations.

Best wishes,

The authors

Reviewers' Comments to Author:

Reviewer: 1

Comments and answers/clarifications:

All in all, the paper and the idea behind it are very compelling. However, there are a few things that can be improved upon.

Thank you for the encouraging comment. We hope we managed to improve the paper under the guidance provided, to make a more reliable conceptual backing and better interpret the findings.

In the introduction/literature review - have you considered establishing a link between the research and the resource-based view of strategy design (as described by Barney, Rumelt, Schmalensee, Grant, and others)? The prevailing idea of strategic management and value management in today's environment is that it should be based on the resources, knowledge, skills, and experience of analysts, planners, and managers. This connection could broaden your research from a theoretical standpoint since you are in effect investigating how KIBS are coping with the strategic shock (as described by Mintzberg) of the COVID-19 pandemic, and how these firms are reactively/proactively trying to deter risks and exploit opportunities in a highly complex and turbulent environment. All of this might help you create a more effective conclusion.

This suggestion is much welcomed and we integrated it into the introductory section to better set the conceptual context of the investigation. Please, see page 3 in the manuscript.

The research design is something that should be explained in more detail, especially the hypotheses, which are missing.

Thank you for the suggestion. We were under the pressure to respect the recommended length so probably we were too brief in explaining the methodology. We added more details on pages  7 and 8. being an explorative research and because only a very few previous studies have been dedicated to this topic, we did not have in mind a set of hypotheses.

The discussion of the results is not coherent, which reflects in the conclusion.

This is a valuable insight and we tried to correct the mentioned aspects both by rethinking the logic of the discussions section and enhancing the conclusion section with additional aspects, including implications.

Reviewer 2 Report

Abstrakt

In the abstract, it must be emphasized that there were a total of 4 local KIBS companies from Romania and semi-structured interviews were conducted with 16 selected employees from these companies. Otherwise, it feels misleading and evokes 16 companies. The abstract should also include the central objective or research questions.

Introduction

The introduction is generally written without content logic, i.e. from a general introduction to the issue to a specific focus with regard to the researched area. I would also welcome the results of the survey of the KM concept in the area of awareness, management approach and adoption (e.g. Poland, Hungary, Slovakia, Bulgaria, Romania, etc.) it is obvious that more than 70% of companies use an informal approach and rather self-serving, often operative management knowledge. That´s nothing new...

The introduction should contain the research question and research objective. It is a standard procedure in qualitative research design as well.

In other words, why do I only find out about the goal of the study somewhere towards the end of the literature search? ".............paper focuses on analyzing the ?specific? challenges determined by the COVID-19 pandemic in relation to Knowledge Management in KIBS in the ??specific??? situation of Romania. A sample of four leading diverse Romanian KIBS has been investigated.

The research questions are only mentioned in the methodology: What are the new challenges for knowledge management, caused by the COVID-19 pandemic? How did the pandemic influence knowledge management performance, including cooperation with customers?

Research questions (RQ´s) help grasp the theoretical framework for literary research and deeper understanding in areas or, on the contrary, point out a gap or the impossibility of grasping the issue with standard tools. Key areas for literature research based on RQ´s: knowledge management process (KIBS - internal, intercompany/ co-creation process), knowledge management performance (KMP), influence of pandemic on KMP, challenges caused by the COVID-19.

Paragraph: "Their business models are credited with more dynamism compared to other types of businesses, as well as high adaptability [5]. "... This leads to a series of challenges [6]." - What challenges? And what do the authors mean by this?

In the introduction, the sector should not be defined in the entire specification and description of NACE Rev 2 (a matter of methodology, specifically the sample). It stretches the text unnecessarily.

Instead, the authors should have focused on survey-based data from a statistical portal in Romania and briefly evaluated the development of these services during the crisis. Perhaps they would find that KIBS are somewhat of a stabilizing factor of economy, all the more the less they are directly dependent on industry....

Literature review

In my opinion, the description of the issue of knowledge management within the literature search is insufficient. It is rather a superficial statement, instead of a specification of the type of knowledge (external/internal, explicit, tacit, implicit) and KM as a process including acquisition, creation, identification, capturing, collection, organization, application, sharing, transferring and distributing (three broad dimensions emerge: knowledge acquisition, knowledge conversion and knowledge application), which will improve the learning process and the performance of the organization. Which of these dimensions does your research focus on?

Materials and Methods

I praise the authors for choosing a qualitative research design, a method of non-standardized questioning using the technique of semi-structured interviews, taking into account the nature and depth of information they need to obtain in order to answer the research questions.

In my opinion, the authors should have pointed out the emergence of new and expanded KIBS support services in literary research (cyber protection, etc.), and not just here.

What do the authors mean in terms of qualitative sampling: "four leading organization" - market share criteria? market coverage in Romania? It is not transparent. What does mean " well-respected organizations"?

I praise for the description of the implementation of the interviews, but nowhere do I see how the data was analyzed and reduced for the needs of the research. Verbatim transcription? Qualitative content analysis? What else was used for coding? What tools did you use to interpret the results and visualize them (verbatim transcription, summary protocol, selective protocol, table, mind map, diagram)? Did you use any software for processing qualitative data (eg Atlas.ti, MaxQda, NViVo), or was everything done "by hand"?

Representatives of professional and technological KIBS were selected, why does the study not also consider "creative" KIBS, as the authors refer to Miles' typology in introduction?

RESULTS

Unfortunately, the results show an inconsistency in the understanding and concept (description) of the KM process, which is a consequence of the level of processing of the literature search. In the description of companies, the authors combine formation and informal approach with cultural elements, areas of KM dimensions, different types of tools, which seems chaotic and confusing with regard to contrasting results. 

Chapter 3.2

The content has great potential for the informative value of the problems faced by KIBS in Romania during Covid-19, but unfortunately it can be seen that the authors do not have much experience in the analysis and interpretation of qualitative data. The text would also deserve to be lightened and accompanied by the support of tables, where the differences, etc., would be indicated. I also recommend separating the text with subsections between intra-company KM and inter-company KM (or via KM dimension´s, approaches). 

3.3. Performance and customer relationship during the pandemic

The content of this chapter speaks rather about the need to maintain relationships based on virtual teams or online (digital) communication and platforms. In fact, the declarative value of KM performance is low and unstructured, even though it is an informal approach to measurement by informants. 

Discussion

The discussion gives answers to the posed research questions and fulfills the research objective. So I wonder where the answer is to:

What are the new challenges for KM, caused by the COVID-19 pandemic? (accompany the text with a supporting table or visual diagram that clearly indicates what new challenges (problems) are involved, e.g. opportunities, limitations-gaps, etc.).

How did the pandemic influence KM performance? It does not matter whether it is an informal approach to KM in companies. From the results of studies of the KM concept in the area of awareness, management approach and adoption (e.g. Poland, Hungary, Slovakia, Czech Republic, Bulgaria, Romania, etc.) it is obvious that more than 70% of companies use an informal approach and rather self-serving, often operative management knowledge than incorporating it into the corporate strategy and yet they are successful.

How did the pandemic influence cooperation with customers?

I recommend dividing the discussion in this way and in the form of subsections.

Among other things, the results of previous researches, which were described in the literature search as pivotal and supporting for the subsequent analysis and interpretation of the data, are compared here. In the discussion, I see the authors' attempt to interpret the results based on the study conducted by Abualqumboz, based on qualitative research of technology-based KIBS in the UK, which pointed to three phases to accommodate the challenges determined by the pandemic: disharmony, normalization, and harmony. Or am I mistaken?

Among other things, the final discussion interprets the results achieved in terms of their possible application in practice, possibly in the context of starting points for possible recommendations for interest groups.

RESULTS

Unfortunately, the results show an inconsistency in the understanding and concept (description) of the KM process, which is a consequence of the level of processing of the literature search. In the description of companies, the authors combine formation and informal approach with cultural elements, areas of KM dimensions, different types of tools, which seems chaotic and confusing with regard to contrasting results. 

Chapter 3.2

The content has great potential for the informative value of the problems faced by KIBS in Romania during Covid-19, but unfortunately it can be seen that the authors do not have much experience in the analysis and interpretation of qualitative data. The text would also deserve to be lightened and accompanied by the support of tables, where the differences, etc., would be indicated. I also recommend separating the text with subsections between intra-company KM and inter-company KM (or via KM dimension´s, approaches). 

3.3. Performance and customer relationship during the pandemic

The content of this chapter speaks rather about the need to maintain relationships based on virtual teams or online (digital) communication and platforms. In fact, the declarative value of KM performance is low and unstructured, even though it is an informal approach to measurement by informants. 

Discussion

The discussion gives answers to the posed research questions and fulfills the research objective. So I wonder where the answer is to:

What are the new challenges for KM, caused by the COVID-19 pandemic? (accompany the text with a supporting table or visual diagram that clearly indicates what new challenges (problems) are involved, e.g. opportunities, limitations-gaps, etc.).

How did the pandemic influence KM performance? It does not matter whether it is an informal approach to KM in companies. From the results of studies of the KM concept in the area of awareness, management approach and adoption (e.g. Poland, Hungary, Slovakia, Czech Republic, Bulgaria, Romania, etc.) it is obvious that more than 70% of companies use an informal approach and rather self-serving, often operative management knowledge than incorporating it into the corporate strategy and yet they are successful.

How did the pandemic influence cooperation with customers?

I recommend dividing the discussion in this way and in the form of subsections.

Among other things, the results of previous researches, which were described in the literature search as pivotal and supporting for the subsequent analysis and interpretation of the data, are compared here. In the discussion, I see the authors' attempt to interpret the results based on the study conducted by Abualqumboz, based on qualitative research of technology-based KIBS in the UK, which pointed to three phases to accommodate the challenges determined by the pandemic: disharmony, normalization, and harmony. Or am I mistaken?

Among other things, the final discussion interprets the results achieved in terms of their possible application in practice, possibly in the context of starting points for possible recommendations for interest groups.

CONCLUSION

The conclusion should synthesize the most important findings of the previous parts of the article. The conclusion therefore contains a brief description of the entire article and the most important findings. Why am I only learning about what "gap" in the literature the authors are reacting to and where the added value and contribution of these results is (it should have been mentioned already in the introduction). After that, research questions.....

I recommend separating the content of the text discussing the limits of the research from the conclusion.

The paper meets the basic standards and requirements for processing an article using the IMRAD method and logic.

Author Response

Dear Reviewer, 

We are very grateful for the detailed feedback and the multitude of constructive recommendations. We hope we could well cope with them and managed to include them in a clear and satisfactory manner. 

See below, for details related to how they have been incorporated in, hopefully, more relevant and valuable research. 

PS: We highlighted in red our clarifications

Reviewer: 2

Comments and answers/clarifications:

Abstrakt

In the abstract, it must be emphasized that there were a total of 4 local KIBS companies from Romania and semi-structured interviews were conducted with 16 selected employees from these companies. Otherwise, it feels misleading and evokes 16 companies. The abstract should also include the central objective or research questions.

Thank you for signaling the ambiguous character of the abstract. We hope we corrected it while managing to keep it concise.

Introduction

The introduction is generally written without content logic, i.e. from a general introduction to the issue to a specific focus with regard to the researched area. I would also welcome the results of the survey of the KM concept in the area of awareness, management approach and adoption (e.g. Poland, Hungary, Slovakia, Bulgaria, Romania, etc.) it is obvious that more than 70% of companies use an informal approach and rather self-serving, often operative management knowledge. That´s nothing new...

The introduction should contain the research question and research objective. It is a standard procedure in qualitative research design as well.

In other words, why do I only find out about the goal of the study somewhere towards the end of the literature search? ".............paper focuses on analyzing the ?specific? challenges determined by the COVID-19 pandemic in relation to Knowledge Management in KIBS in the ??specific??? situation of Romania. A sample of four leading diverse Romanian KIBS has been investigated.

The research questions are only mentioned in the methodology: What are the new challenges for knowledge management, caused by the COVID-19 pandemic? How did the pandemic influence knowledge management performance, including cooperation with customers?

Research questions (RQ´s) help grasp the theoretical framework for literary research and deeper understanding in areas or, on the contrary, point out a gap or the impossibility of grasping the issue with standard tools. Key areas for literature research based on RQ´s: knowledge management process (KIBS - internal, intercompany/ co-creation process), knowledge management performance (KMP), influence of pandemic on KMP, challenges caused by the COVID-19.

Paragraph: "Their business models are credited with more dynamism compared to other types of businesses, as well as high adaptability [5]. "... This leads to a series of challenges [6]." - What challenges? And what do the authors mean by this?

In the introduction, the sector should not be defined in the entire specification and description of NACE Rev 2 (a matter of methodology, specifically the sample). It stretches the text unnecessarily.

Instead, the authors should have focused on survey-based data from a statistical portal in Romania and briefly evaluated the development of these services during the crisis. Perhaps they would find that KIBS are somewhat of a stabilizing factor of economy, all the more the less they are directly dependent on industry....

We found all suggestions useful, therefore we tried to integrate all of them. We enhanced the theoretical framework, presented in a more focused way the research questions and design, we excluded the NACE classification which indeed did not bring really added-value to the discourse. We believed we also improved the readability of this section.

Literature review

In my opinion, the description of the issue of knowledge management within the literature search is insufficient. It is rather a superficial statement, instead of a specification of the type of knowledge (external/internal, explicit, tacit, implicit) and KM as a process including acquisition, creation, identification, capturing, collection, organization, application, sharing, transferring and distributing (three broad dimensions emerge: knowledge acquisition, knowledge conversion and knowledge application), which will improve the learning process and the performance of the organization. Which of these dimensions does your research focus on?

The need to enhance the conceptual background of the research was very valuable to us. Therefore, we added some clarification regarding the knowledge-based view of a company and the structure of knowledge. We gave more details on processes specific to KM that are going to be later discussed for the Romanian KIBS in the findings section.

We also developed the literature review of the existing investigation, especially since we identified two more papers on the Romanian market. Overall, we added almost 20 new relevant references that might help readers have a more consistent view of the extant research.

Materials and Methods

I praise the authors for choosing a qualitative research design, a method of non-standardized questioning using the technique of semi-structured interviews, taking into account the nature and depth of information they need to obtain in order to answer the research questions.

In my opinion, the authors should have pointed out the emergence of new and expanded KIBS support services in literary research (cyber protection, etc.), and not just here.

What do the authors mean in terms of qualitative sampling: "four leading organization" - market share criteria? market coverage in Romania? It is not transparent. What does mean " well-respected organizations"?

I praise for the description of the implementation of the interviews, but nowhere do I see how the data was analyzed and reduced for the needs of the research. Verbatim transcription? Qualitative content analysis? What else was used for coding? What tools did you use to interpret the results and visualize them (verbatim transcription, summary protocol, selective protocol, table, mind map, diagram)? Did you use any software for processing qualitative data (eg Atlas.ti, MaxQda, NViVo), or was everything done "by hand"?

Representatives of professional and technological KIBS were selected, why does the study not also consider "creative" KIBS, as the authors refer to Miles' typology in introduction?

We are aware that many aspects can be detailed and additionally discussed for a complex understanding of the methodological approaches and their implementation. We tried to keep all these aspects focused to meet the formal requirements of the editors and leave more space for discussing the theoretical background, the findings, and their relevance.  We tried to correct all the aspects mentioned while also being synthetic. We hope we were successful in implementing relevant, yet brief, clarifications as constructively suggested.  

RESULTS

Unfortunately, the results show an inconsistency in the understanding and concept (description) of the KM process, which is a consequence of the level of processing of the literature search. In the description of companies, the authors combine formation and informal approach with cultural elements, areas of KM dimensions, different types of tools, which seems chaotic and confusing with regard to contrasting results.

Chapter 3.2

The content has great potential for the informative value of the problems faced by KIBS in Romania during Covid-19, but unfortunately it can be seen that the authors do not have much experience in the analysis and interpretation of qualitative data. The text would also deserve to be lightened and accompanied by the support of tables, where the differences, etc., would be indicated. I also recommend separating the text with subsections between intra-company KM and inter-company KM (or via KM dimension´s, approaches).

3.3. Performance and customer relationship during the pandemic

The content of this chapter speaks rather about the need to maintain relationships based on virtual teams or online (digital) communication and platforms. In fact, the declarative value of KM performance is low and unstructured, even though it is an informal approach to measurement by informants.

Thank you for the observation related to the difficulty of following the presentation of the results. We structured it considering your feedback, dividing the presentation into KM processes and practices. We also included a few synthetical tables for easier following of the main aspects revealed. New focus has been put on the functional solutions, in most cases, that companies adopted to the new challenges. Section 3.3. is consistent with the findings, which, as observed, are not very complex in connection with this topic. It seems that KM is a consistent concern, as well as customer relationships, but not many thoughts are given to measuring and controlling all these aspects. This helped us extend the implications-related discussions. This new approach of the section also helped us better design discussions and conclusions, so please also follow those sections in connection with this one.

Discussion

The discussion gives answers to the posed research questions and fulfills the research objective. So I wonder where the answer is to:

What are the new challenges for KM, caused by the COVID-19 pandemic? (accompany the text with a supporting table or visual diagram that clearly indicates what new challenges (problems) are involved, e.g. opportunities, limitations-gaps, etc.).

How did the pandemic influence KM performance? It does not matter whether it is an informal approach to KM in companies. From the results of studies of the KM concept in the area of awareness, management approach and adoption (e.g. Poland, Hungary, Slovakia, Czech Republic, Bulgaria, Romania, etc.) it is obvious that more than 70% of companies use an informal approach and rather self-serving, often operative management knowledge than incorporating it into the corporate strategy and yet they are successful.

How did the pandemic influence cooperation with customers?

I recommend dividing the discussion in this way and in the form of subsections.

Among other things, the results of previous researches, which were described in the literature search as pivotal and supporting for the subsequent analysis and interpretation of the data, are compared here. In the discussion, I see the authors' attempt to interpret the results based on the study conducted by Abualqumboz, based on qualitative research of technology-based KIBS in the UK, which pointed to three phases to accommodate the challenges determined by the pandemic: disharmony, normalization, and harmony. Or am I mistaken?

Among other things, the final discussion interprets the results achieved in terms of their possible application in practice, possibly in the context of starting points for possible recommendations for interest groups.

We hope we could take into account all aspects mentioned. We also believe we improved the readability of this section by re-organizing the discourse. We also included an explicit recommendation for the managers section at the end of the paper.

CONCLUSION

The conclusion should synthesize the most important findings of the previous parts of the article. The conclusion therefore contains a brief description of the entire article and the most important findings. Why am I only learning about what "gap" in the literature the authors are reacting to and where the added value and contribution of these results is (it should have been mentioned already in the introduction). After that, research questions.....

I recommend separating the content of the text discussing the limits of the research from the conclusion.

The paper meets the basic standards and requirements for processing an article using the IMRAD method and logic.

Thank you again for all the valuable insights and recommendations that guided us through the process of improving the paper. We also considered these last suggestions, hopefully respecting their spirit. We clarified various ideas, enhanced the limits section, and we introduced an implication section.

Round 2

Reviewer 2 Report

Abstract

The abstract already has a better narrative value in relation to the key areas.

Introduction

I appreciate the effort of the authors to respond to the recommendations in the field of brief description of the state and development of KIBS in Romania. National statistical offices do not record statistical data for this "type" of services. As a rule, it is either based on own analyzes according to NACE Rev. 2 or the European Commission and others (EUROSTAT, European commission studies or OECD) regularly inform about KIBS in (Eastern) Europe. This is also used by your colleagues, researchers: BUSU, Cristian; I WILL, Mihail. The role of knowledge intensive business services on Romania's economic revival and modernization at the regional level. Sustainability, 2017, 9.4: 526. CÄ‚TOIU, Iacob; TUDOR, Livius; BISA, Cristian. Knowledge-intensive business services and business consulting services in the Romanian changing economic environment. Amphitheater Economic Journal, 2016, 18.41: 40-54. BÄ‚DULESCU, Daniel; SIMUT, Ramona; HERTE, Anamaria Diana. Linking Kibs, Entrepreneurial Dynamics And Macroeconomic Developments. Focus On Romania. In: Proceedings of the INTERNATIONAL MANAGEMENT CONFERENCE. Faculty of Management, Academy of Economic Studies, Bucharest, Romania, 2018. p. 202-212.

The most recent data for KIBS in Romania (2000 - 2018) is presented for example in the article: Cristina Zeldea, 2019. "Knowledge-Intensive Business Services. The Case Of Romania," Revista de Economie Mondiala / The Journal of Global Economics, Institute for World Economy, Romanian Academy, vol 11 (1), pages 54-58.

Otherwise, the logic of the issue is better structured in the introduction.

Literature review

Although the literature review is improved with regard to the connection of RBV, deliberate strategy and other studies, it is still not entirely clear which areas of the KM process you are focusing on. Could you visualize it? I.e. examining aspects related to the organization's KM strategy, its KM processes and KM performance (simple diagram or mind map?). That would help a lot. KM performance is therefore "hidden" in the part of the chapter "How important is KM for KIBS" at the end. Or am I mistaken?

I recommend removing the paragraph: "Having in mind the uncovered evolutions of KIBS knowledge management.......(...)"... categories of KIBS (t/p/c KIBS)" from the theoretical part. The given argument for relevance and focus should have been used in the introduction and methodologically supported in the methods.

Materials and methods

Improvements have been made in the description of data collection and analysis methods (including the description of the purposive sample of interviewees).

Results

I greatly appreciate the effort to visualize the results in the form of a table (table 2). It is necessary to refer in the text to what it actually displays. This is not an inter-case comparison, but rather a synthesis of similarities or the occurrence of characteristics across the investigated processes. For me, it would be appropriate to carry out the synthesis in a discussion, however....ok.

A big shift for the better took place within the processing of chapter 3.2. I would call Chapter 3.2.1 "New challenges of KM processes". Again, to increase professionalism, I recommend using visualization of key results.

Chapter 3.2.2 is almost completely reworked and for the better! Finally, the practices are better described and, even better, it is clear from Table 3 that this was related to tactical changes (solutions).

Chapter 3.3 is much less developed compared to the results areas mentioned above. It is necessary to realize what KM performance, even if it is informal, corresponds to in terms of level (strategic, tactical, operative? or can we divide it in terms of the performance of individual processes?). This is not clear from the entire article. You are talking about improving performance in these areas (indicators):internal: improvement of knowledge flows in the organization and between employees, efficiency of adaptation of new employees...........external: effective or improvement of customer relations, communication with a customers, better customer service, building and realizing projects, improvement of obtaining industry specific knowledge.....

Digital performance evaluation map is a tool....again, aspect (indicator) - solution?

I miss the visualization of key results here.

I would place the "customer relations" part in the previous chapter as "challenges or gaps" rather than performance outcomes.

Discussion

In the discussion, I would welcome the display of key results such as overall linkage, looking at aspects of (KM strategy) KM processes, KM challenges (gaps) and KM performance or the visualization of the results: "three phases of their adaptation to the challenges caused by the pandemic: disharmony, normalization and harmony.".

In the discussion, I would welcome the display of key results, i.e. knowledge-gaps (challenges) as a synthesis.

Conclusion - ok

Chapter 5.1 and 5.2 improved and modified according to comments.

I recommend specifying the title of the article. The current title: "Managing knowledge in KIBS during the COVID-19 pandemic" is too general.

I recommend for example: 1) Knowledge Management in Romanian KIBS during Covid-19 pandemic or 2) Knowledge Management in KIBS during Covid-19 pandemic in Romania

Author Response

Dear editors and reviewers,

Thank you again for the thorough review associated with our manuscript entitled "Managing knowledge in KIBS during the COVID-19 pandemic". We appreciate very much your encouragement, appreciation, and the useful feedback that contributes to the improvement of the paper.

Please see below (in red) our responses to your concerns, in round 2 of the review process.

We will upload two versions of the revised manuscript – one with track changes to easier follow the modifications and one without track changes for a clearer view of the present form of the research. We hope we could correctly grasp and incorporate your recommendations.

Best wishes,

The authors

Comments and answers/clarifications:

Title

I recommend specifying the title of the article. The current title: "Managing knowledge in KIBS during the COVID-19 pandemic" is too general.

I recommend for example 1) Knowledge Management in Romanian KIBS during the Covid-19 pandemic or 2) Knowledge Management in KIBS during Covid-19 pandemic in Romania

We incorporated the suggestion to be more specific.

Abstract

The abstract already has a better narrative value in relation to the key areas.

Thank you for the confirmation

Introduction

I appreciate the effort of the authors to respond to the recommendations in the field of brief description of the state and development of KIBS in Romania. National statistical offices do not record statistical data for this "type" of services. As a rule, it is either based on own analyzes according to NACE Rev. 2 or the European Commission and others (EUROSTAT, European commission studies or OECD) regularly inform about KIBS in (Eastern) Europe. This is also used by your colleagues, researchers: BUSU, Cristian; I WILL, Mihail. The role of knowledge-intensive business services on Romania's economic revival and modernization at the regional level. Sustainability, 2017, 9.4: 526. CĂTOIU, Iacob; TUDOR, Livius; BISA, Cristian. Knowledge-intensive business services and business consulting services in the Romanian changing economic environment. Amphitheater Economic Journal, 2016, 18.41: 40-54. BĂDULESCU, Daniel; SIMUT, Ramona; HERTE, Anamaria Diana. Linking Kibs, Entrepreneurial Dynamics And Macroeconomic Developments. Focus On Romania. In: Proceedings of the INTERNATIONAL MANAGEMENT CONFERENCE. Faculty of Management, Academy of Economic Studies, Bucharest, Romania, 2018. p. 202-212.

The most recent data for KIBS in Romania (2000 - 2018) is presented for example in the article: Cristina Zeldea, 2019. "Knowledge-Intensive Business Services. The Case Of Romania," Revista de Economie Mondiala / The Journal of Global Economics, Institute for World Economy, Romanian Academy, vol 11 (1), pages 54-58.

Otherwise, the logic of the issue is better structured in the introduction.

Thank you for the valuable additional references. We have analyzed them, as well as the European innovation scoreboard, and we have deepened the section on the Romanian KIBS highlighting better its characteristics and impact on the economy.

Literature review

Although the literature review is improved with regard to the connection of RBV, deliberate strategy, and other studies, it is still not entirely clear which areas of the KM process you are focusing on. Could you visualize it? I.e. examining aspects related to the organization's KM strategy, its KM processes, and KM performance (simple diagram or mind map?). That would help a lot. KM performance is, therefore "hidden" in the part of the chapter "How important is KM for KIBS" at the end. Or am I mistaken?

I recommend removing the paragraph: "Having in mind the uncovered evolutions of KIBS knowledge management.......(...)"... categories of KIBS (t/p/c KIBS)" from the theoretical part. The given argument for relevance and focus should have been used in the introduction and methodologically supported in the methods.

Developing a mind map is a very good idea for succinctly imagining the investigated universe of KM. We included such an image, focused only on the dimensions investigated in the current research.

The highlighted paragraph has been moved to the beginning of the methodological section.

Materials and methods

Improvements have been made in the description of data collection and analysis methods (including the description of the purposive sample of interviewees).

Thank you for the positive feedback.

Results

I greatly appreciate the effort to visualize the results in the form of a table (table 2). It is necessary to refer to the text to what it actually displays. This is not an inter-case comparison, but rather a synthesis of similarities or the occurrence of characteristics across the investigated processes. For me, it would be appropriate to carry out the synthesis in a discussion, however....ok.

A big shift for the better took place within the processing of chapter 3.2. I would call Chapter 3.2.1 "New challenges of KM processes". Again, to increase professionalism, I recommend using visualization of key results.

Chapter 3.2.2 is almost completely reworked and for the better! Finally, the practices are better described and, even better, it is clear from Table 3 that this was related to tactical changes (solutions).

Chapter 3.3 is much less developed compared to the results areas mentioned above. It is necessary to realize what KM performance, even if it is informal, corresponds to in terms of level (strategic, tactical, operative? or can we divide it in terms of the performance of individual processes?). This is not clear from the entire article. You are talking about improving performance in these areas (indicators):internal: improvement of knowledge flows in the organization and between employees, the efficiency of adaptation of new employees...........external: effective or improvement of customer relations, communication with customers, better customer service, building and realizing projects, improvement of obtaining industry-specific knowledge.....

Digital performance evaluation map is a tool....again, aspect (indicator) - solution?

I miss the visualization of key results here.

I would place the "customer relations" part in the previous chapter as "challenges or gaps" rather than performance outcomes.

Thank you for your appreciation and the further suggested improvements. We have included the following aspects:

  • a brief comment on the content of Table 2
  • we changed the title of section 3.2.1 as suggested.
  • we kept together the KM performance and customer relationship aspects because, as also explained in the text, the respondents measure the performance of their KM in connection with customer relationships. It is the most important aspect revealed in the case of all companies. We added some discussions at the end of section 3.3 to stress once more this aspect, and also to answer the issues raised by your comments.

Discussion

In the discussion, I would welcome the display of key results such as overall linkage, looking at aspects of (KM strategy) KM processes, KM challenges (gaps) and KM performance or the visualization of the results: "three phases of their adaptation to the challenges caused by the pandemic: disharmony, normalization and harmony.".

In the discussion, I would welcome the display of key results, i.e. knowledge-gaps (challenges) as a synthesis.

We tried to make discussions clearer by inserting various comments and highlighting more critical aspects.

Conclusion - ok

Chapter 5.1 and 5.2 improved and modified according to comments.

Thank you for your support and patience to go through the paper so attentively and provide detailed comments and recommendations.
